# Advancing Transgender Health Education for Health Professional Students: A Faculty Fellowship Model for Capacity Building and Curricular Change

**DOI:** 10.3390/healthcare13172124

**Published:** 2025-08-26

**Authors:** Whitney R. Linsenmeyer, Katie Heiden-Rootes, Laura Burch, Genevieve Del Rosario, Katie Sniffen, Ashley D. Schmuke, Kristi Richter, Michelle R. Dalton, Rabia Rahman, Theresa Drallmeier, Rowan Hollinger

**Affiliations:** 1Department of Nutrition and Dietetics, Saint Louis University, Saint Louis, MO 63103, USA; rabia.rahman@health.slu.edu (R.R.); rowan.hollinger@slu.edu (R.H.); 2Department of Family and Community Medicine, School of Medicine, Saint Louis University, Saint Louis, MO 63103, USA; katie.heidenrootes@health.slu.edu (K.H.-R.); michelle.dalton@health.slu.edu (M.R.D.); theresa.drallmeier@slucare.ssmhealth.com (T.D.); 3Department of Psychology, Saint Louis University, Saint Louis, MO 63103, USA; laura.burch1@health.slu.edu; 4Department of Clinical Health Sciences, Saint Louis University, Saint Louis, MO 63103, USA; genevieve.delrosario@health.slu.edu; 5Department of Physical Therapy and Athletic Training, Saint Louis University, Saint Louis, MO 63103, USA; katie.sniffen@health.slu.edu; 6Trudy Busch Valentine School of Nursing, Saint Louis University, Saint Louis, MO 63103, USA; ashley.schmuke@slu.edu; 7School of Social Work, Saint Louis University, Saint Louis, MO 63103, USA; kristi.richter@slu.edu

**Keywords:** transgender, health education, curriculum

## Abstract

**Background**: Promoting the health of transgender and gender diverse people requires healthcare providers who are competent in gender-affirming communication and care. However, transgender health education is often absent in health professions curricula or isolated to a single course. This is partly attributed to faculty feeling unprepared to train students on key concepts in transgender health. **Purpose**: The purpose of this short communication is to describe the development and implementation of a novel Faculty Fellowship in Transgender Health, the outcome of which is a curricular map of integrated transgender health content in various healthcare professions. Five faculty members participated in the inaugural fellowship from the disciplines of physician assistant, nursing, athletic training, social work, and clinical psychology programs. **Recommendations**: By providing resources, training, and a network of peer educators, institutions can support faculty to think holistically about content relevant to transgender health and potential curricular adaptations in their respective disciplines.

## 1. Introduction

Transgender patients are poorly served by the United States (US) healthcare system. Nearly one in four (24%) transgender adults did not see a doctor when they needed to in the last year due to fear of mistreatment [1]. Among those who did see a healthcare provider in the past year, nearly one in two (48%) reported at least one negative experience related to their gender, such as being refused healthcare, being misgendered, having a provider use harsh or abusive language towards them, or having a provider be physically rough or abusive [1]. Lack of training in transgender health for health professional students drives the negative experiences of transgender patients [2,3]; patients reported anticipating highly negative experiences with healthcare providers, avoiding healthcare altogether, and having to educate their providers on aspects of their own care [4]. Other challenges persist related to insurance barriers and state-level restrictions on gender-affirming care [1,5].

### 1.1. Educational Interventions Are Often Limited

Sexual and gender minority-health related content is not required for medical school accreditation or medical student training [6,7]. Nonetheless, many educational programs have successfully modeled the efficacy of brief interventions to improve student knowledge, attitudes, beliefs, comfort, and confidence in caring for transgender clients [2,3,8,9]. Most programs have utilized lectures as an educational intervention to address this gap. Other programs included simulations, videos or films, guest speakers, reading assignments, reflective writing assignments, guided discussions, or problem-based learning, among others [2,8]. Most educational interventions have been situated within a single course [2,3] or in individual clinical supervision with trained supervisors [10]. Few studies have modeled an interprofessional education approach [11,12], despite relevant skills that intersect all healthcare professions. The lack of long-term follow-up data and varied nature of interventions further limit existing research [9]. While research supports that brief educational interventions, including as little as a one-hour lecture, may effectively improve student learning outcomes specific to transgender health [11,12], students still desire more training to feel competent in providing gender-affirming care [13].

Novel approaches have sought to integrate transgender health content throughout a curriculum, such as the Transgender Curriculum Integration Project which focused on five courses in an accelerated Bachelor’s in Nursing Science program [12,13]. Others have created an optional certificate in lesbian, gay, bisexual and transgender (LGBT) Health [14]. These developments are promising but are limited to a single discipline or are optional to students. Infusing transgender health content throughout curricula is necessary to better prepare health professionals to provide gender-affirming care for the future patients and clients [15,16].

### 1.2. Faculty Feel Unprepared to Deliver Transgender Health Content

Frequently cited reasons for the absence of transgender health content are that faculty feel unprepared, time-constrained by other workload demands, or under-qualified to adequately address relevant topics [2,8,17,18,19,20,21]. Other barriers include limited curricular time, poor institutional support, or a perception of irrelevance [8]. Importantly, state-level legislation banning diversity, equity, and inclusion (DEI) efforts at public institutions [22] will require faculty to think critically and creatively about how to prepare their students to provide culturally competent care for diverse populations. Thus, efforts toward faculty development, competency, and institutional support are warranted [8,17].

Scholars have studied faculty readiness [19,20] and emphasized the need for robust faculty development programs to sustainably integrate transgender health into curricula [2,8,14,15]. However, these recommendations are largely informed by the expertise of scholars in transgender health. Descriptions of actual faculty development programs, especially for faculty with little knowledge in transgender health, are largely lacking. As one example, the Transgender Curriculum Integration Project briefly described the process for contacting nursing faculty members, soliciting agreement their agreement to participate, and coaching them on how to adopt new content in their courses [14]. Detailed, robust descriptions of faculty development approaches are necessary ensure this important recommendation can be actualized in health education.

### 1.3. Current Recommendations for Health Professional Student Education in Transgender Health

Current recommendations for student education in transgender health emphasize the importance of integrating content throughout curricula to widely address critical topics that may naturally fit in various courses, such as gender-affirming communication, clinical assessment, and gender identity development [15,17,18,23,24]. This approach has been modeled by scholars in the fields of nursing [14,15,24], medicine [17], and athletic training [25], among others. By thinking holistically about their curricula, educators can consider how to best address relevant content at appropriate points in their curricula and integrate a range of biological, psychosocial, and clinical perspectives.

The current literature also emphasizes the value of fostering interprofessional collaboration among faculty seeking to adapt their curricula [2,17]. Faculty champions may emerge organically as they collaborate with transgender health experts in other disciplines, clinical partners, and community-based organizations [2]. Strategies to cultivate faculty collaboration include developing a health education interest directory, making informal introductions among individuals or groups with a shared interest in transgender health, or issuing institutional calls for participating in relevant projects [17].

### 1.4. Purpose

The purpose of this short communication is to describe the development of a Faculty Fellowship in Transgender Health and to disseminate an exemplar curricular map of transgender health content across multiple healthcare professions.

## 2. Faculty Fellowship in Transgender Health: Development and Implementation

The Transgender Health Collaborative at Saint Louis University is a network of faculty researchers, educators, and clinicians working with the transgender community throughout Saint Louis University and community organizations. Our core team consists of non-binary and cisgender faculty from marital and family therapy, medicine, and nutrition and dietetics.

We developed a Faculty Fellowship in Transgender Health designed for educators in a broad spectrum of health professions, including allied health, mental health, and behavioral health disciplines. The purpose was to support faculty members in healthcare disciplines seeking to advance the inclusion of transgender and gender diverse identities within their courses and across their curricula.

Five faculty members were invited to participate in the fellowship program. We elected to invite faculty to represent a range of disciplines and who had previously worked with the Transgender Health Collaborative at Saint Louis University and therefore were likely to be enthusiastic about the opportunity. We also sought to include faculty who were serving in leadership roles given the recommendation to catalyze interest from curriculum developers [17]. The five faculty members were from the physician assistant, nursing, athletic training, social work, and clinical psychology programs. Three fellows were also program directors (physician assistant, athletic training, social work), and one was a director of clinical services (clinical psychology). Faculty fellows were remunerated with a stipend. The fellowship was supported through a grant from the Josiah Macy Jr. Foundation.

### 2.1. Four Components of the Faculty Fellowship in Transgender Health

The Faculty Fellowship in Transgender Health consisted of four components throughout the fall semester (Figure 1). The fellowship was limited to one semester given the challenge of working with time-constrained faculty and the subsequent recommendation to avoid overburdening individuals [17].

First, we provided fellows with a curated reading list that included selected publications from the field of transgender health more broadly, as well as those germane to their disciplines. All fellows were assigned to read My Child is Trans, Now What? A Joy-Centered Approach to Support by Ben Greene, a St. Louis-based author and educator. Fellows were also assigned to read a resource authored by the Transgender Health Collaborative at Saint Louis University titled “Advancing Inclusion of Transgender & Gender Diverse Identities in Clinical Education: A Toolkit for Clinical Educators” [23]. The remaining assigned readings included position statements, commentaries, and additional learning resources (Table 1).

Second, we hosted a 90 min lunch discussion where we debriefed on the readings and reflected on challenges and opportunities to better integrate transgender health content in our curricula (Table 2). Third, all fellows participated in the 2024 Interprofessional Transgender Health Education Day. This is a full-day training on introductory concepts in transgender health and gender-affirming communication and care. The learning objectives, session descriptions, and outcomes of this training have been reported previously [33].

Fourth, we hosted a second 90 min lunch workshop where we discussed what we learned from the Interprofessional Transgender Health Education Day and began developing a curricular map of how we could better integrate transgender health content in our respective curricula (Table 3). We also incorporated our own disciplines (nutrition and dietetics and marital and family therapy) into the work as an opportunity to improve our own curricula and work alongside the faculty fellows. Faculty were given three additional weeks to complete the curricular mapping after the workshop, the outcomes of which are detailed in the next section.

### 2.2. Integration of Concepts in Transgender Health Throughout Health Professions Curricula

Appendix A depicts the curricular map of key concepts in transgender health in our respective disciplines. The Fellows first determined the key concepts in transgender health that should be reflected in a health professions curriculum, which were also informed by the current literature. These included seven concepts: Gender identity development [17]; gender minority populations [14,15]; historical and social context of transgender people [8,14,15,17]; legal, policy, and ethical implications for transgender people [8,14,15]; counseling and communication skills with transgender clients [14,17]; practicum and simulation with transgender patients or clients [8]; and research methods and sex, sexual orientation, and gender identity (SOGI) data [17]. The seven broad concepts invited an intersectional approach to consider a range of life experiences such as race, ethnicity, education, income level, and social class [34]. For example, the concept of “historical and social context of transgender people” could be applied to educate students on differences within the transgender population, such as the higher rates of poverty, homelessness, and violence experienced by transgender people of color [1].

Fellows were challenged to identify courses, learning objectives, readings, and learning activities for each key concept (Appendix A). They were encouraged to think about a curriculum in their field more broadly, rather than the specific program in place for their discipline at Saint Louis University, to support generalizability of the curricular map to various institutions. Fellows were also encouraged to focus only on their ideas for the ideal content, rather than the logistics of who, when, and how the content would be operationalized.

## 3. Discussion and Future Directions

The Fellows successfully identified multiple courses in their disciplines to deliver each key concept (Appendix A). For example, the concept of gender identity development could be taught in a Lifespan Development and/or a Human Diversity Course within a clinical psychology program. This underpins the flexible nature of how curricula may be adapted at different institutions according to faculty strengths, time, and desire. In addition, the Fellows identified readings and learning activities that could be readily utilized by other healthcare disciplines. For example, a proposed learning activity from the nursing fellow involved critically evaluating a nursing research study to assess how the researchers collected SOGI data and identifying potential improvements in data collection practices. This exercise could be adapted for any healthcare discipline by using an article relevant to their field. Thus, even if an academic program was not prepared to adapt their entire curriculum, components could still be utilized on a smaller scale, such as new readings, case studies, or learning objectives in a single course.

Successful implementation of the curricular map will require stakeholder buy-in from the faculty and administration in the respective programs. Faculty members would need to make changes to their courses such as adding or modifying the course objectives, incorporating new assignments and reading materials, and developing evaluation methods for new assignments [14]. This requires both curricular time and faculty time, which are known challenges to incorporating transgender health content [17]. A potential risk is that some faculty members would readily make changes to their courses, while others would not, thus creating a possible fracturing in how the content is presented cohesively and consistently across the curriculum. This risk could be ameliorated by clear and tangible support from university leaders, such as provosts, deans, department chairs, and program directors, as well as accrediting bodies who actively influence the curriculum. Notably, curricular adaptation on a smaller scale, such as one or two courses with updated reading materials, would still be an improvement from a dearth of transgender health content.

In addition, successful implementation would require faculty to update content somewhat frequently given the rapidly evolving nature of transgender health research [35]. Faculty support would be needed to ensure the content is up to date, such as access to high quality continuing education, subscription to relevant journals in the university library, and connections to peer colleagues engaged in similar work [2,17]. While continuing education requirements of all healthcare professionals create opportunities to obtain up-to-date transgender health information, providers may feel challenged to balance this need with other ongoing field-specific requirements for continuing education credits (CEUs).

Successful implementation would also require approaches to ensure stability of the content in the curriculum, which may be challenged by faculty turnover or reassignment to different courses. Realistically, when a complete curricular change is unrealistic or unsupported, content may still be championed by one or two faculty members. Methods could include a “warm hand-off” from one faculty member to the next with the sharing of readings, assignments, or guest speakers, integration of the content within program learning outcomes that may be regularly assessed and reported, and clear alignment with competencies determined by the accrediting body of health profession education programs.

Lastly, faculty expertise around transgender healthcare may become clustered within certain departments of healthcare professions at teaching institutions, as faculty may naturally collaborate more frequently with those within their discipline or department. Scholars have recommended strategies to foster intellectual exchange and collaboration related to transgender health more broadly, such as institution-wide participation calls, informal introductions, or a transgender health education interest directory [17]. In our experience, the formation of a “collaborative,” or essentially, a network of faculty formally recognized by the university, has led to rich and interdisciplinary collaboration in our research, teaching, and clinical practice [36]. The interprofessional nature of the Faculty Fellowship in Transgender Health allowed expert faculty to provide training across professional and institutional divides, thus strengthening trainee education in transgender healthcare for the institution as a whole.

### Limitations

This short communication describes the development and implementation of a novel program rather than a traditional research study. As such, inherent limitations included the lack of measurable outcomes; future research may explore outcomes related to faculty experiences, teaching evaluations, and student reception of new content. Furthermore, this fellowship was implemented at a single institution with five faculty members and therefore may not be generalizable to institutions of other sizes, regions, or public versus private affiliation. Institutions may need to consider what adaptations would make a faculty fellowship program successful at their unique institution.

## 4. Conclusions

Development and implementation of a Faculty Fellowship in Transgender Health was an important capacity-building step towards better incorporating transgender health in various curricula. By providing resources, training, and a network of peer educators, faculty were supported to think holistically about relevant content and curricular adaptations. While challenges will persist such as faculty buy-in and turnover, this model offers a promising approach to build faculty competency and foster collaboration. This fellowship required relatively limited time commitments from faculty across only one semester, making it more feasible for faculty participation. As this short communication describes development and implementation of a new program, future research can systematically evaluate outcomes of interest, such as faculty experiences and student learning. Ultimately, integration of transgender health content throughout curricula is essential to ensure that the next generation of healthcare professionals are equipped to deliver excellent care for their future patients and clients.

## Figures and Tables

**Figure 1 healthcare-13-02124-f001:**
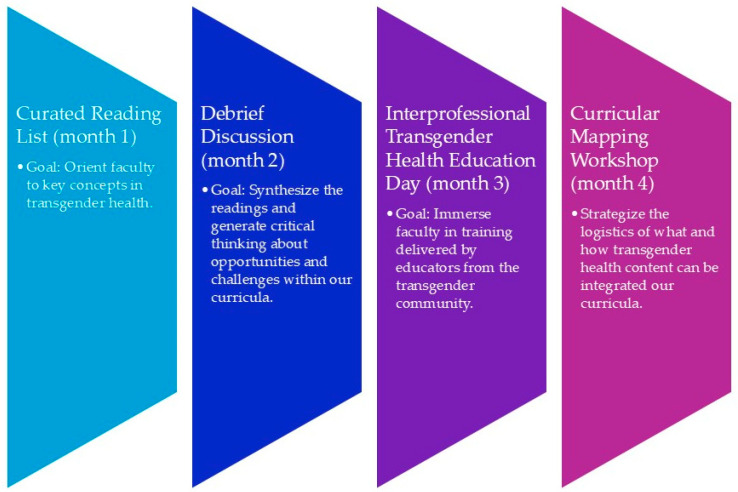
Four elements of the faculty fellowship in transgender health.

**Table 1 healthcare-13-02124-t001:** Reading list for five faculty fellows.

Discipline	Reading 1	Reading 2
Physician Assistant	Curricular Approaches to Transgender Health in Physician Assistant Education [21]	Collection of resources from the National LGBTQIA + Health Education Center [26]
Nursing	Transgender Health Care for Nurses: An Innovative Approach to Diversifying Nursing Curricula to Address Health Inequities [14]	American Nursing Association (ANA) Position Statement: Nursing Advocacy for LGBTQ + Populations [27]
Athletic Training	The Role of the Athletic Trainer in Providing Care to Transgender and Gender-Diverse Patients: Foundational Knowledge and Disparities–Part I [28]	The Role of the Athletic Trainer in Providing Medical Care to Transgender and Gender-Diverse Patients: Considerations for Medical Affirmation–Part II [29]
Clinical Psychology	American Psychological Association (APA) Policy Statement on Affirming Evidence-Based Inclusive Care for Transgender, Gender Diverse, and Nonbinary Individuals, Addressing Misinformation, and the Role of Psychological Practice and Science [30]	Collection of resources from the National LGBTQIA + Health Education Center [26]
Social Work	Collection of resources from the National Association of Social Workers [31]	Collection of resources from the National LGBTQIA + Health Education Center [26]

Note. All five fellows were also assigned to read My Child is Trans, Now What? A Joy-Centered Approach to Support by Ben Greene [32] and “Advancing Inclusion of Transgender & Gender Diverse Identities in Clinical Education: A Toolkit for Clinical Educators” [23].

**Table 2 healthcare-13-02124-t002:** Debrief discussion guide for the faculty fellowship in transgender health.

Introductions	Name, pronouns Icebreaker
Review of Readings	What stood out as surprising or inspiring from what you have read so far? What questions remain about transgender healthcare or transgender people and experiences?
Discussion	How do you see a gender-affirming perspective fitting into the culture of your field and/or department? How does your curriculum currently address sex, gender, gender diversity, and/or transgender health, if at all? What knowledge, skills, or competencies do graduating students need to possess in order to provide excellent gender-affirming care for their future transgender patients/clients?
Dreaming	Who in your network may be a champion of this work at Saint Louis University? Who may be a challenge? (Don’t need to name folks—just think about different stakeholders)

**Table 3 healthcare-13-02124-t003:** Discussion guide for the curricular mapping workshop of the faculty fellowship in transgender health.

Introductions	Name, pronouns Icebreaker
Reflection from the Interprofessional Transgender Health Education Day	What moments stick out with you the most? How did conversations go with your students before/after the event? What should we keep/change for next year?
Discussion/Workshop	What core concepts in transgender health should be included in a health professional curriculum? What courses in your discipline would be an ideal place to address each concept? What learning objectives, assignments, and readings can be used to introduce or reinforce each core concept?
Next Steps	Manuscript submission Stipends

## Data Availability

No new data were created or analyzed in this study.

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
