# Peer review of "Advancing Transgender Health Education for Health Professional Students: A Faculty Fellowship Model for Capacity Building and Curricular Change"

_healthcare, 2025, doi:10.3390/healthcare13172124_

Round 1

Reviewer 1 Report

Comments and Suggestions for Authors

Thank you for the opportunity to review this important and timely manuscript focused on building institutional capacity for transgender health education through a faculty fellowship model. The topic is highly relevant, especially as healthcare education seeks to advance inclusive and equitable curricula across disciplines.

Overall, the manuscript is well-structured, clearly written, and grounded in both scholarly literature and practical application. The interdisciplinary approach and the development of a curricular map are valuable contributions that can serve as a model for other institutions.

However, I offer the following suggestions to improve the clarity and completeness of the manuscript:

Ethics Declarations:
Although the project focuses on faculty development and curriculum design, the manuscript is missing the required statements on institutional review board (IRB) approval and informed consent. Even if not applicable, standard wording should be included (e.g., “Ethical review and approval were waived…”).

Methodological Design:
The manuscript would benefit from a clearer explanation of the methodology. For instance, how were participants selected? Were any tools or data collection methods used to evaluate the fellowship’s outcomes? If no formal data was collected, please clarify this explicitly.

References and Citations:
The reference list is generally appropriate, but the introduction would benefit from inclusion of a few more recent and comprehensive studies or reviews that document the state of transgender health education across disciplines. Additionally, placeholder references marked as “BLINDED” should be replaced with full citations prior to final submission.

Introduction Enhancements:
In addition to strengthening references, consider elaborating further on why faculty development—specifically—addresses the gap in transgender health education, as this is central to your contribution.

Language and Expression:
While generally well-written, some portions of the manuscript could benefit from careful editing to enhance clarity and fluency. A professional language revision or internal peer editing may be helpful prior to final submission.

Author Contributions:
One of the contributor fields lists “X.X.” — please ensure all author initials are correctly matched and verified.

Figures and Tables:
All tables and figures are informative and well-designed. Please ensure they meet the resolution requirements (600 dpi) for final publication.

This manuscript is well positioned for publication once these minor issues are addressed. I commend the authors for their innovative and collaborative approach to advancing transgender health education and look forward to seeing the final version.

Author Response

Please see the attachment for the point-by-point response. Thank you for your thorough review and feedback. 

Reviewer 2 Report

Comments and Suggestions for Authors

This is a descriptive communication piece, not a research study, so I did not look for scientific soundness, etc.. It is clearly written and easy to read. It has bias, but then champions of causes need to have biases.

I think its potential is that it offers practical suggestions that can be used for teaching. The authors point out some of the difficulties of implementing the program in its entirety, so that is a strength of the paper. Given the realities of the obstacles discussed, I would like to have seen recommendations from the authors on how parts of the program could be used and still be beneficial. I think the authors would see more uptake of their work that way. As it stands, the only option seems to do the entire program and that could cause readers to not even finish reading the paper. Of course, the authors might not want to do this. The decision has to be whether having part of the program delivered is better than nothing being done.

It is well-written and seems to be a thoughtfully developed program. It is worth publishing.

Author Response

(The authors gave the same response as above.)

Reviewer 3 Report

Comments and Suggestions for Authors

attached file

Author Response

(The authors gave the same response as above.)

Reviewer 4 Report

Comments and Suggestions for Authors

Promoting health education for students in health professions: a scholarship model for teachers for training and curriculum change.
Today, training in transgender health is not exclusive to the field of medicine or medical sciences, but also to the social sciences (social work and psychology).
The proposed text is in the form of a communication. The text is relevant but needs to be rewritten in a style suitable for an article and requires the introduction of a point on the method used to arrive at this systematisation. It also needs a discussion on gender theories and how this proposal relates to them.

Author Response

(The authors gave the same response as above.)

Reviewer 5 Report

Comments and Suggestions for Authors

Dear authors,

Comgratulations on your work.

Please revise your manuscript according to MDPI rules:

"Free Format Submission

Healthcare now accepts free format submission:

  • We do not have strict formatting requirements, but all manuscripts must contain the required sections: Author Information, Abstract, Keywords, Introduction, Materials & Methods, Results, Conclusions, Figures and Tables with Captions, Funding Information, Author Contributions, Conflict of Interest and other Ethics Statements. Check the Journal Instructions for Authors for more details."

Author Response

(The authors gave the same response as above.)

Round 2

Reviewer 1 Report

Comments and Suggestions for Authors

The revised manuscript meets the quality criteria for publication in this esteemed journal.

Author Response

Please see the attached table. 

Reviewer 2 Report

Comments and Suggestions for Authors

I read the edits and they are sufficient to justify publication. I think this is a paper with merit.

Author Response

Please see the attached table. 

Reviewer 4 Report

Comments and Suggestions for Authors

The authors improved the initial text, resulting in a greater contribution to knowledge.

Author Response

Please see the attached table. 

Reviewer 5 Report

Comments and Suggestions for Authors

Dear authors:

Congratulations on the improvement made on your article. Even so, I do not believe it meets the required standards of HealthCare.

I would suggest a new view on this article, trying to highlight why it is important and why it brings something new to what already is known. Another take on the chosen type of paper or methodology would help to achieve those goals, in my opinion.

Best regards.

Author Response

Please see the attached table. 
